# Deep Lipschitz networks and Dudley GANs

## Abstract

Generative adversarial networks (GANs) have enjoyed great success, however often suffer instability during training which motivates many attempts to resolve this issue. Theoretical explanation for the cause of instability is provided in Wasserstein GAN (WGAN), and wasserstein distance is proposed to stablize the training. Though WGAN is indeed more stable than previous GANs, it takes more iterations and time to train. This is because the ways to ensure Lipschitz condition in WGAN (such as weight-clipping) significantly limit the capacity of the network. In this paper, we argue that it is beneficial to ensure Lipschitz condition as well as maintain sufficient capacity and expressiveness of the network. To facilitate this, we develop both theoretical and practical building blocks, using which one can construct different neural networks using a large range of metrics, as well as ensure Lipschitz condition and sufficient capacity of the networks. Using the proposed building blocks, and a special choice of a metric called Dudley metric, we propose *Dudley* GAN that outperforms the state of the arts in both convergence and sample quality. We discover a natural link between Dudley GAN (and its extension) and empirical risk minimization, which gives rise to generalization analysis.

## 1 Introduction

Generative adversarial networks (GANs) (Goodfellow et al., 2014) are recently proposed powerful generative models that use two neural networks: (1) a generator network that produces fake samples that tries to look as real as possible; and (2) a discriminator network that seeks to distinguish fake and real samples.

The discriminator typically utilizes a *divergence measure* between the distributions of the reals and fakes. Early works of GANs are shown to correspond to Jensen-Shannon divergence (Goodfellow et al., 2014), and later other divergence measures are introduced to GANs such as *f-divergences* (Nowozin et al., 2016). These divergence based GANs often suffer from training instability, and several heuristic tricks have been used to tackle this issue such as batch normalization and careful network initialization. Arjovsky et al. (2017) point out at theoretical level that the use of divergence has several flaws such as infinite divergence and vanishing gradient when the two distributions have no common supports (intersection measure zero). They propose to use earth mover's distance which does not need common support. Directly minimizing the earth mover's distance can be expensive, thus they resort to solve its dual *wasserstein measure* instead. They use several tricks such as weight clipping (Arjovsky et al., 2017) and constrain the norm of the gradient (Gulrajani et al., 2017) to ensure Lipschitz condition in neural nets. However, these significantly limit the capacity of the network and reduce the convergence speed.

Ensuring Lipschitz condition in neural nets are the true cornerstone of stablizing GANs. GANs are sensible to Lipschitz constant: a small value leads to a simpler function with low capacity however a larger value can be unstable to train. This is why bounded Lipschitz functions are desirable. Neural networks use complex functions composed of layered nonlinearities, thus analysis of the Lipschitz condition is highly nontrivial. *This is why we develop building blocks to facilitate easy construction of new GANs that always ensure Lipschitz condition.*

With the proposed building blocks, and Dudley metric, we propose a new *Dudley* GAN which outperforms the state-of-the-arts due to ensuring both Lipschitz condition and sufficient network capacity. We further show that it can be extended to a class of metrics called *integral probability*

*metric* (IPM) (Müller, 1997; Sriperumbudur et al., 2012), of which wasserstein measure (used in WGAN (Arjovsky et al., 2017; Gulrajani et al., 2017)), maximum mean discrepancy (used in various GANs (Mroueh et al., 2017; Karolina Dziugaite et al., 2015; Sutherland et al., 2016; Lim & Ye, 2017)) and Dudley metric are special cases. Viewing GANs in IPM, we can view our approach as minimizing a "function norm" subject to best discrimination.

Our contributions in this paper are as follows: (1) we provide sufficient conditions and building blocks for a Lipschitz neural net (Section 2); (2) we introduce Dudley GAN and show utilizing minimum bounded Lipschitz functions with appropriate regularization for GANs are effective in producing good quality images with stable training and faster convergence (Section 3); (3) we show that our Dudley GAN (and its extension) has a natural link to regularized empirical risk minimization, and provide a generalisation bound on error of the discriminator (Section 4) using Rademacher complexity.

## 2 DEEP LIPSCHITZ NETWORKS

Lipschitz condition is crucial for GANs, and for neural networks in general, here we provide sufficient conditions for a deep neural network to be Lipschitz.

### 2.1 THEORETICAL ASPECT

A function $f$ is *Lipschitz* (or Lipschitz continuous) on a space $\mathcal{X}$ if there exists a constant $K \geq 0$ such that $\|f(\mathbf{x}_1) - f(\mathbf{x}_2)\| \leq K\|\mathbf{x}_1 - \mathbf{x}_2\|$ for all $\mathbf{x}_1, \mathbf{x}_2 \in \mathcal{X}$. $K$ is known as a *Lipschitz constant*. Similar conditions can be found in other communities, e.g. control theory (Fromion, 2000).

**Definition 1** (Lipschitz semi-norm). Lipschitz semi-norm of a function $f$ on a space $\mathcal{X}$ is

$$\|f\|_L := \sup\left\{\frac{|f(\mathbf{x}_1) - f(\mathbf{x}_2)|}{|\mathbf{x}_1 - \mathbf{x}_2|} : \mathbf{x}_1 \neq \mathbf{x}_2 \forall \mathbf{x}_1, \mathbf{x}_2 \in \mathcal{X}\right\}.$$

$\|f\|_L$ is in fact the smallest Lipschitz constant of $f$ (also known as *the* Lipschitz constant for short). This means that every differentiable function with bounded first derivative is Lipschitz with Lipschitz constant $\|f\|_L = \sup |f'(\mathbf{x})|$, and a function is Lipschitz when its Lipschitz semi-norm is defined everywhere.

**Definition 2.** [Lipschitz Neural Network] A neural network represents a function $f_\mathbf{w} : \mathcal{X} \to \mathcal{Y}$ that maps from input space $\mathcal{X}$ to output space $\mathcal{Y}$ parameterized by a set of parameters $\mathbf{w}$. We say the neural network is a Lipschitz Neural Network if $\|f_\mathbf{w}\|_L$ is defined everywhere in its input space.

By Definition 2, we know that a single activation function such as Perceptron is a Lipschitz neural network if that activation function is Lipschitz.

**Proposition 1.** For two Lipschitz neural nets $f_{\mathbf{w}_1}$ and $g_{\mathbf{w}_2}$ parameterized by $\mathbf{w}_1, \mathbf{w}_2$ with Lipschitz constant $L_f$ and $L_g$ respectively, we have:

(i) linear combination $h_\mathbf{w}(\mathbf{x}) = f_{\mathbf{w}_1}(\mathbf{x}) + g_{\mathbf{w}_2}(\mathbf{x})$ is a Lipschitz neural network with constant $L_f + L_g$ and a new set of parameters denoted by $\mathbf{w}$;

(ii) the composition function $h_\mathbf{w}(\mathbf{x}) = f_{\mathbf{w}_1}(g_{\mathbf{w}_2}(\mathbf{x}))$ is Lipschitz neural network with constant $L_f \times L_g$ with a new set of parameters denoted by $\mathbf{w}$.

Proposition 1 shows that a *deep* neural network defined by a composition of functions is a Lipschitz neural network if the functions in all layers are Lipschitz. And the Lipschitz constant of the deep neural network is the product of the Lipschitz constants of the functions in all layers.

**Proposition 2.** Applying following operations in a Lipschitz neural network preserves Lipschitz:

(i) Convolution is an operation that satisfies Lipschitz condition on a compact space with Lipschitz constant $K$ when $|\mathbf{w}| = K$ for $d$-dimensional[1] weight $\mathbf{w}$ or $Kd$ where $\|\mathbf{w}\| = K$.

(ii) A linear function with weight vector $\mathbf{w}$ where $\|\mathbf{w}\| = K$ satisfies Lipschitz condition with Lipschitz constant $K$.

---

[1] In practice, $d = 1 + (w - F)/s$ where $s$ is the number of strides, $w$ is the input volume and $F$ is the receptive field (kernel size) of the input.

(iii) Following nonlinear activation functions: Sigmoid, $\tanh$ and Rectified Linear Unit (ReLU), exponential linear unit (ELU), sinusoid ($\sin$), softplus defined as $\frac{1}{\beta}\log(1 + \exp(\beta x))$.

(iv) $\log(x)$ where $x \in [1, \infty)$.

There are other operations in neural networks such as batch and weight normalization (as linear transformations from a compact space to another) and dropout that are widely used and preserve Lipschitz.

## 2.2 IMPLEMENTING DEEP LIPSCHITZ NEURAL NETWORKS

Above properties provide guidance for designing Lipschitz Neural Networks, and can be implemented in practice. Here we discuss a few practical tricks can be used or have been used, and relate them to the properties.

**Weight Normalization** (Salimans & Kingma, 2016a): A simple reparameterization trick for linear and convolutional layers, shown to be extremely effective in practice. It is an efficient approach for ensuring the weights are normalized and the network is Lipschitz. It should be noted that the learnable scale parameter (equaling the norm of parameter vector $\|\mathbf{w}\|$), is not guaranteed to be constant throughout training. As such, the Lipschitz constant of the network is learned.

$L_2$-**Regularization**: Another approach is to penalize the weight values with large $L_2$ norm. It provides a soft constraint on the norm of the weight vector.

**Batch normalization** (Salimans & Kingma, 2016b): Another approach for normalizing the inputs so that they are all normally distributed with mean zero. The mean and standard deviation is optimized during training and is shown to perform well as a regularizer. For Lipschitz neural networks, it ensures the output of the batch normalized layer is bounded which effectively constrains the weights and gradients. In practice, Xiang & Li (2017) showed weight normalization is more effective for GANs than batch normalization.

**Gradient norm regularization and clipping**: Since Lipschitz condition limits the rate of change in functions corresponding to the gradient in continuous functions, it is reasonable to consider regularizing the gradient. In general we don't need any gradient regularization as the Lipschitz neural nets ensure bounded gradients. However, in practice this value may be large and incur sudden changes in the network that can lead to divergence. Clipping the gradient is simple however might lead to divergence because it changes the direction in which the parameters of the network are updated. A better choice is to scale the gradients by their norm when the gradients are too large. We found this scaling can be effective in practice when the data is too complex and batches in Stochastic Gradient Descent (SGD) optimization vary which cause sudden changes in the gradient.

Weight normalization in combination with $L_2$ regularization enable parameters to be updated smoothly when used with variations of SGD in back-propagation. Additional $L_2$ regularization ensures new parameters of weight normalization are also regularized. van Laarhoven (2017) provides an extensive analysis of the combination of these two.

## 3 DUDLEY GANS

With the guidance of the properties presented and a choice of a special metric called Dudley metric, we develop a new GAN called Dudley GAN, which outperforms the state of the arts in both convergence and sample quality, due to ensured Lipschitz and sufficient network capacity. We then show that Dudley metric belongs to a broader class of metrics that can be integrated nicely with GANs. This opens a door to create new GANs with desirable theoretical and practical properties.

## 3.1 BOUNDED DEEP LIPSCHITZ NEURAL NETWORKS FOR GANS

Let $p$ be the true data distribution where the real samples are drawn from, and $q_{\boldsymbol{\theta}}$ be the fake data distribution represented by the generator network parametrised by $\boldsymbol{\theta}$. We only have access to the real samples but not $p$. Generating a fake sample $\mathbf{x}' \sim q_{\boldsymbol{\theta}}$ is typically done by drawing a random vector $\mathbf{z}$ from a simple and known distribution $p_{\mathbf{z}}$ first, and then feed $\mathbf{z}$ into the generator network

which outputs the fake same, that is $\mathbf{x}' = g_{\boldsymbol{\theta}}(\mathbf{z})$. Here $g_{\boldsymbol{\theta}}$ is a deterministic function that maps the generator network's input to its output. In the ideal case, $q_{\boldsymbol{\theta}}$ and $p$ are "equal" with respect to the divergence measure of choice (that is the distance becomes zero). The discriminator utilizes the divergence measure and provides feedback to the generator.

Let $f_{\mathbf{w}}$ be the function that maps the input of the discriminator to its output with parameter $\mathbf{w}$. We propose to use *integral probability measure* (Müller, 1997) below for the generator:

$$\rho(p, q_{\boldsymbol{\theta}}) \quad = \quad \sup_{f_{\mathbf{w}} \in \mathcal{F}} \left| \int_{\mathcal{X}} f_{\mathbf{w}} dp - \int_{\mathcal{X}} f_{\mathbf{w}} dq_{\boldsymbol{\theta}} \right|, \tag{1}$$

where $\mathcal{F}$ is a class of real-valued measurable bounded functions on $\mathcal{X}$. Setting $\mathcal{F} = \{ f_{\mathbf{w}} : \|f_{\mathbf{w}}\|_{BL} \leq 1 \}$ in Equation 1 where $\|.\|_{BL}$ is defined as $\|f_{\mathbf{w}}\|_{BL} = \|f_{\mathbf{w}}\|_{\infty} + \|f_{\mathbf{w}}\|_{L}$ with $\|f_{\mathbf{w}}\|_{\infty} = \sup \{ |f_{\mathbf{w}}(\mathbf{x})| : \mathbf{x} \in \mathcal{X} \}$ produces a metric called *Dudley metric* (Müller, 1997). Effectively, this metric constrains the function to be both bounded and Lipschitz. It is worth noting that Dudley metric is a weaker convergence measure compared to the Wasserstein distance. However, for GANs weaker measure is more desirable as it deters the discriminator from saturation (Liu et al., 2017). We consider the following cases of bounded Lipschitz functions detailed below.

**Constrained Linear**: We set the function $f_{\mathbf{w}}$ to be both Lipschitz and bounded in the output. As such we have the following objective:

$$\max \qquad \mathbb{E}_{\mathbf{x} \sim p}[f_{\mathbf{w}}(\mathbf{x})] - \mathbb{E}_{\mathbf{x}' \sim q_{\boldsymbol{\theta}}}[f_{\mathbf{w}}(\mathbf{x}')], \qquad \text{subject to } -\gamma \leq f_{\mathbf{w}}(\mathbf{x}) \leq \gamma \qquad \forall \mathbf{x}, 0 < \gamma < 1$$

Alternatively, this problem can be written as the following unconstrained objective

$$\rho_{L_2}(p, q_{\boldsymbol{\theta}}) \quad = \quad \max \min_{\lambda} \quad \mathbb{E}_{\mathbf{x} \sim p}[f_{\mathbf{w}}(\mathbf{x})] - \mathbb{E}_{\mathbf{x}' \sim q_{\boldsymbol{\theta}}}[f_{\mathbf{w}}(\mathbf{x}')] - \lambda R(\mathbf{w}), \quad R(\mathbf{w}) = (\|\mathbf{f}_{\mathbf{w}}\| - \gamma)^2, \tag{2}$$

where we have a soft constraint with $\lambda$ as the Lagrangian multiplier. Here, $\mathbf{f}$ is the vector constructed from the samples of $p$ and $q_{\boldsymbol{\theta}}$. In fact we approximate the $\|.\|_{\infty}$ constraint for the functional norm with the $L_2$ norm of the function values for samples from both distributions. At each step of the optimization, in addition to the parameters of the discriminator and generator, we need to update the Lagrangian multiplier.

**Using $\tanh$**: Another alternative is to use a function with limited output while the network is Lipschitz. We use $\tanh$ as a popular bounded activation function, i.e.

$$\rho_{\tanh}(p, q_{\boldsymbol{\theta}}) \quad = \quad \max \quad \mathbb{E}_{\mathbf{x} \sim p}[f_{\mathbf{w}}(\mathbf{x})] - \mathbb{E}_{\mathbf{x}' \sim q_{\boldsymbol{\theta}}}[f_{\mathbf{w}}(\mathbf{x}')], \qquad f_{\mathbf{w}}(\mathbf{x}) := \tanh(h_{\mathbf{w}}(\mathbf{x}))$$

where $h_{\mathbf{w}}$ is a Lipschitz neural network mapping the input $\mathbf{x}$ to a real number. With $\tanh$ as the output of the network, we might suffer from the vanishing gradient. Considering the derivative of $\tanh$, that is $1 - \tanh(x)^2$ for a real value $x$, the gradient is at its peak at $\tanh(x) = 0$. When the output is close to $\pm 1$ for either real or fake samples, the gradient approaches zero and as such there would be no more improvement on the discriminator or more importantly on the generator. As such, we directly constrain the value of the function to be close to zero. In particular, we use

$$R(\mathbf{w}) \quad = \quad (\|\mathbf{f}_{\tanh}\| - \gamma)^2, \quad , 0 < \gamma < 1, \tag{3}$$

where $\mathbf{f}_{\tanh}$ is the vector of the function values for the samples from the real/fake distribution.

There are other cases that we could consider, however, we observed in practice constraining the rate of improvement with Lipschitz condition as well as bounding the output provides the most stable algorithm for GANs. For instance, using $\tanh$ as the output of WGAN has been tested when the network is not conditioned to be Lipschitz (Gulrajani et al. (2017)) and it is reported that the algorithm diverged (as we further observed in practice).

The optimal generator is obtained when we have:

$$\nabla_{\boldsymbol{\theta}} \rho(p, q_{\boldsymbol{\theta}}) \quad = \quad \nabla_{\boldsymbol{\theta}} \left( \mathbb{E}_{\mathbf{x} \sim p}[f_{\mathbf{w}}(\mathbf{x})] - \mathbb{E}_{\mathbf{z} \sim p_{\mathbf{z}}}[f_{\mathbf{w}}(g_{\boldsymbol{\theta}}(\mathbf{z}))] \right) = -\mathbb{E}_{\mathbf{z} \sim p_{\mathbf{z}}} \left[ \nabla_{\boldsymbol{\theta}} f_{\mathbf{w}}(g_{\boldsymbol{\theta}}(\mathbf{z})) \right] = 0$$

Putting all together, we have our algorithm detailed in Algorithm 1. In this algorithm we use Adam (Kingma & Ba (2014)) that generally is considered a fast and stable optimization.

### 3.2 INTEGRAL PROBABILITY METRICS

Integral probability metrics (IPM) (Müller, 1997; Sriperumbudur et al., 2012) as defined in Equation 1 introduces a powerful class of divergences. Various choices of function class $\mathcal{F}$ lead to well-known divergence measures. Here we provide a few examples.

```
 1: while θ is not converged do
 2:     Sample uniformly x₁, . . . , xₙ from 𝒳
 3:     Sample z′₁, . . . , z′ₙ ∼ p_z
 4:     ρ′_w = ∇_w [ 1/n Σᵢ f_w(xᵢ) − 1/n Σᵢ f_w(g_θ(zᵢ)) ]    {f_w is Linear or tanh}
 5:     R′_w = ∇_w R(w)   {Equation 2 or 3}
 6:     w = w + η_D.Adam(w, ρ′_w − λR′_w)
 7:     Sample z′₁, . . . , z′ₙ ∼ p_z
 8:     ρ′_θ = −∇_θ 1/n Σᵢ f_w(g_θ(zᵢ))
 9:     θ = θ − η_G.Adam(θ, ρ′_θ)
10:     λ = λ − η_λ R′_w
11: end while
```

**Algorithm 1:** The algorithm for the Dudley GAN. In this algorithm, are the learning rates of discriminator, generator and Lagrangian multiplier.

**Wasserstein distance**: Setting $F = \{f : \|f\|_L \leq 1\}$ in Equation 1. The Kantorovich-Rubinstein theorem states that for separable space $\mathcal{X}$, this metric in Equation 1 with Lipschitz condition is a dual for $L_1$-Wasserstein distance. This measure is used in WGAN.

**Total Variation**: Setting $F = \{f : \|f\|_\infty \leq 1\}$ yields popular total variation distance. For example, EBGAN (Zhao et al. (2016)) uses a similar condition.

**Kernel Distance**: Setting $F = \{f : \|f\|_\mathcal{H} \leq 1\}$ yields Maximum Mean Discrepancy (MMD) distance. MMD has been a popular measure in GANs (Karolina Dziugaite et al. (2015); Sutherland et al. (2016); Lim & Ye (2017)).

It should also be noted that when using Lipschitz functions on a compact spaces (such as images) with both positive and negative values, we always have a bounded function, i.e.

$$\|f\|_\infty = \sup_\mathbf{x} |f(\mathbf{x})| \leq \sup_{\mathbf{x},\mathbf{x}'} |f(\mathbf{x}) - f(\mathbf{x}')| \leq \mathrm{diam}(\mathcal{X})\|f\|_L,$$

where $\mathrm{diam}(\mathcal{X})$ is the maximum distance between points in the input. Therefore, while bounding the output does not significantly change the GAN framework, it does provide a practical advantage as well as theoretical (as will be further discussed in Section 4).

We concentrate on bounded Lipschitz neural networks because only using bounded functions (not Lipschitz) for GANs might suffer from the vanishing gradient problem and limit the capacity of the output. Secondly, using MMD typically requires a choice of kernel.

Moreover, it is interesting to note that in an IPM when $f(\mathbf{x}) = \langle \phi(\mathbf{x}), \mathbf{w} \rangle$ for some function $\phi(\mathbf{x})$ and $\|\mathbf{w}\| \leq 1$, we recover MMD. Therefore, for a neural network with linear output where $\phi(\mathbf{x})$ represents one layer to the last and is weight-normalized (no need for the previous layers to be Lipschitz) another alternative for training a GAN is obtained.

## 4 GENERALIZATION BOUNDS

Another benefit of introducing IPM to GANs, is that it provides a natural link between GANs and empirical risk minimization (ERM). If we assign a label $y = +1$ for real samples and $y = -1$ for fake ones, it is easy to see the relation between IPM and ERM as a binary classification when, without loss of generality, we have $p(y = +1) = p(y = -1)$, then

$$\inf_{f \in F} \mathbb{E}_{(\mathbf{x},y)\sim\mu(\mathbf{x},y)} [\ell(y, f_\mathbf{w}(\mathbf{x}))] = \inf_{f \in F} \int \ell(+1, f_\mathbf{w}(\mathbf{x}))\mu(\mathbf{x}|y = +1) + \ell(-1, f_\mathbf{w}(\mathbf{x}))\mu(\mathbf{x}|y = -1)$$

For $\ell(y, f_\mathbf{w}(\mathbf{x})) = -yf(\mathbf{x}), p(\mathbf{x}) = \mu(\mathbf{x}|y = +1), q_\theta(\mathbf{x}') = \mu(\mathbf{x}'|y = -1)$ we recover the objective in Equation 1. Hence, it is easy to see the discriminator's objective is a regularized ERM: $\min_\mathbf{w} \mathbb{E}_{(\mathbf{x},y)\sim\mu(\mathbf{x},y)} [\ell(y, f_\mathbf{w}(\mathbf{x}))] + R(\mathbf{w})$. Although in practice, the complete minimization is not performed in each iteration of GAN.

It can be seen (Sriperumbudur et al., 2012) that the dual of this problem minimizes the norm (depending on the choice of the norm in IPM) of the function at hand, in our case $\|f\|_{BL}$. In other words, we learn the smoothest function (lower $\|f\|_L$) with minimum highest value ($\|f\|_\infty$). We need more *complex functions* (higher Lipschitz constant) when the divergence distance is small (i.e.

distinguishing closer distributions are more difficult). Since in GANs we learn the fake distribution $q_\theta$ it is important to have more complex models for $f_\mathbf{w}$, an indication of mixture of fake and real samples. Thus, constraining the Lipschitz constant to be very small is in fact a deterring factor in estimating the fake distributions from the generalization perspective.

Finally we add that learning the function $f_\mathbf{w}$ is relevant to the *Rademacher complexity* where we sample from the real ($\sigma = 1$) and fake ($\sigma = -1$) distributions with the same probability ($\frac{1}{2}$) and evaluate the empirical Rademacher complexity (Shawe-Taylor & Cristianini, 2004),

$$\mathfrak{R} = \mathbb{E}_\sigma \left[ \sup_{f \in F} \frac{1}{m} \left| \sum_{i=1}^m \sigma_i f(\mathbf{x}_i) \right| \right], \quad \mathbf{x}_i \in \mathcal{X} \cup \mathcal{X}', \mathcal{X}' = \{g(\mathbf{z}_1) \ldots, g(\mathbf{z}_n)\}, \mathbf{z}_i \sim p(\mathbf{z})$$

that measures the random labeling of the data. This simple measure allows us to determine how easy it is for our model to distinguish real from fake samples when we randomly label them. Moreover since our function is bounded, with probability at least $1 - \delta$ for $\delta \in (0, 1)$ over $m$ samples from $\mathcal{X} \cup \mathcal{X}'$, we have a bound on the probability of misclassification of fake/real samples,

$$p[y \neq \text{sgn}(f(\mathbf{x}))] \leq \frac{1}{m\gamma} \sum_{i=1}^m (\gamma - yf(\mathbf{x}))_+ + 4\sqrt{\frac{\text{diam}(\mathcal{X})}{m\gamma^2}} + 3\sqrt{\frac{\log(2/\delta)}{2m}} \tag{4}$$

As observed, the maximum confusion in the discriminator that is desirable for GANs is inversely related to the value of $\gamma$. The proof is provided in the supplements.

## 5 EXPERIMENTS

Examining generative models are generally challenging. In this section, we examine the quality of the generated images using Dudley GAN. To that end, we construct a Lipschitz neural network with the bounded output detailed in Section 2. We build upon the architecture of DCGAN (Radford et al. (2015)) with an additional linear layer on top and weight-normalization in each layer of the discriminator. The network architecture is detailed in Appendix B. We apply batch-normalization in the generator that stabilizes and improves convergence speed. We set the learning rate to $0.0001$, weight decay to $0.001$, batch size $64$ and 100-dimensional noise samples from normal distribution. We, furthermore, compare with gradient clipping where we clip gradient values exceeding value 2 and gradient decay where we normalize the gradients so that its norm is less than 10. For WGAN we set the weight clipping range to $[-0.01, 0.01]$ as is done in the original paper. We will use the same experimental setup throughout the evaluation with various datasets.

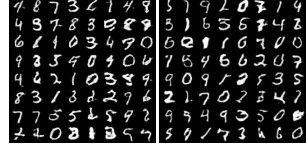

(a) Linear      (b) tanh

Figure 1: Generated samples from MNIST dataset.

For the original GAN and its variants where JS-divergence is used, batch normalization is essential for providing common support for the real and fake datasets. Otherwise, it takes a long time to converge. Batch-norm provides a simple practical trick to encourage high correlation between fake and real samples. When we use WGAN, batch normalization helps to induce a compact space after each layer. In Dudley GAN, batch-norm is not required.

**MNIST Dataset**: This dataset contains $60,000$ hand-written digits formatted as binary $28 \times 28$ images. We use Algorithm 1 to train a generator. Samples from the generator is shown in Figure 1(a) for linear and 1(b) for tanh at iteration $5000$. Moreover, we use kernel density estimation (KDE) to estimate the log-likelihood of the generated samples at iteration $10,000$ similar to Nowozin et al. (2016). As is common

| Algorithm | LL (nats) |
|---|---|
| WGAN (Arjovsky et al., 2017) | $413.27 \pm 0.30$ |
| GAN | $305 \pm 8.97$ |
| tanh-no-weight-norm | $413.31 \pm 6.10$ |
| tanh | $415.32 \pm 1.21$ |
| Bounded Linear | $417.54 \pm 2.1$ |

Table 1: log-likelihood of various approaches for MNIST dataset in nats

practice, we have taken 16K samples from the trained generator and report the mean log-likelihood in Table 5. Even though KDE is not an excellent measure and has high variance, we compared the log-likelihoods of variations approaches. As observed, tanh with Lipschitz network outperforms tanh with no weight normalization (corresponding to the total variation). The performance of the bounded linear function with regularization that is proposed in the paper has the best log-likelihood

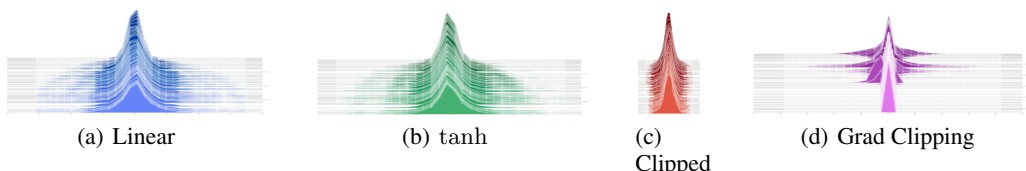

(a) Linear      (b) tanh      (c) Clipped      (d) Grad Clipping

Figure 3: Weight distributions of the last layer of the generator when using weight clipping compare to other cases. As observed, the variance of the weights in the linear and is larger. This implies more diversity in the images produced. In the gradient clipping case where the weight has a very small variance our approach diverges.

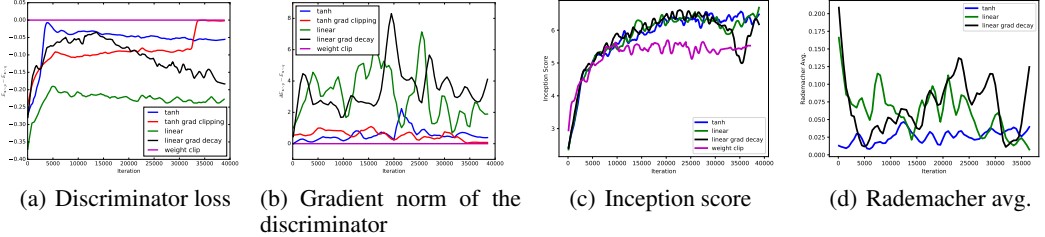

(a) Discriminator loss    (b) Gradient norm of the discriminator    (c) Inception score    (d) Rademacher avg.

Figure 4: CIFAR-10 training metrics

albeit with larger variance. Lower variance in WGAN is due to the extremely restrictive nature of weight clipping.

**CIFAR-10 Dataset**: The CIFAR-10 dataset Krizhevsky & Hinton (2009) is composed of $50,000$ RGB images of natural scenes with size $32 \times 32$. We train our GAN similar to MNIST except that here we are using three channels. The progress of the generator is shown in Figure 2. It is interesting to observe that weight clipping exhibits relatively fast initial improvement however our approach catches on quickly. This is because as the time passes the capacity of the discriminator is exploited and the frequency of the weights being clipped increases. As such, the rate of improvement in the network significantly decreases. In addition, when we use tanh for the output of a Lipschitz network, initially the improvement of the generator is slow however we observe a steady improvement and the end result exhibit better diversity and sharper quality compared to WGAN.

In Figure 4 we show metrics from training of the CIFAR-10 dataset. In particular, in Figure 4(a) the discriminator loss without the regularization is shown. As observed, tanh and bounded linear function show a stable value for the difference in Equation 1 (empirical expected functions). However, when we use other regularizations, namely gradient clipping or gradient normalization we observe the discriminator loss is not stable. In addition, in Figure 4(b) the norm of the gradient of the loss is plotted and as is observed generally gradient of the tanh is smaller than the bounded linear. This is because as the output of tanh increases the value of its gradient decreases, which is not the case with the linear function. One interesting case is when we use gradient clipping, the algorithm diverges which is be-

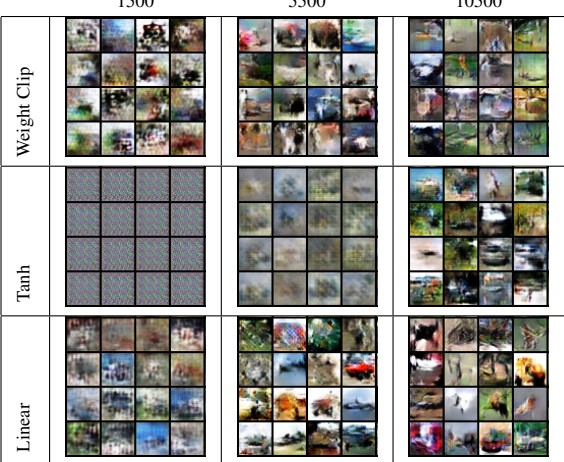

Figure 2: Generated images at various steps of training. As observed weight clipping saturates very fast.

cause the crude change of the gradient alters discriminator in a wrong way. In Figure 4(c), we use Inception score (Salimans et al., 2016) to evaluate the visual features of the generated images. Even though this is not a comprehensive measure, we observe our approach performs better than WGAN (and WGAN-GP). Furthermore, Rademacher complexity is also shown in Figure 4(d). The functional value of the tanh is smaller and thus its Rademacher value, however the final value for the bounded linear is smaller.

In Figure 3 we show the distribution of the weights in the generator. As observed, when using weight clipping in the discriminator, the output of the generator is also very restricted. As such, the range

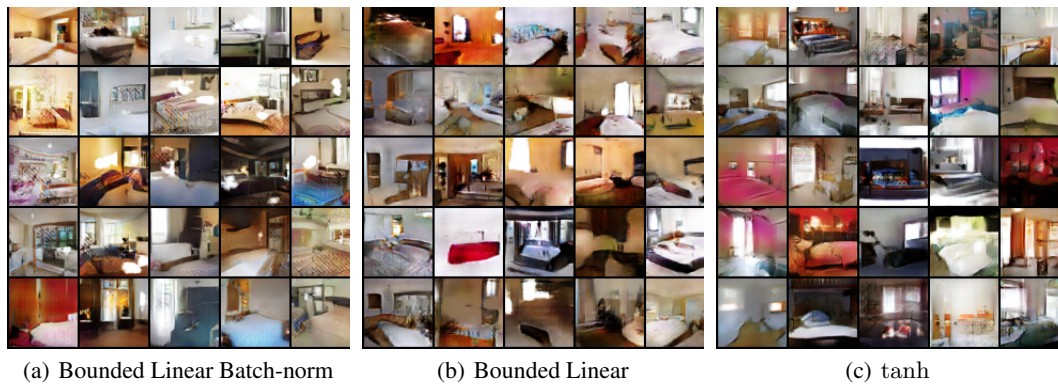

| (a) Bounded Linear Batch-norm | (b) Bounded Linear | (c) tanh |

Figure 5: Images generated from the LSUN dataset.

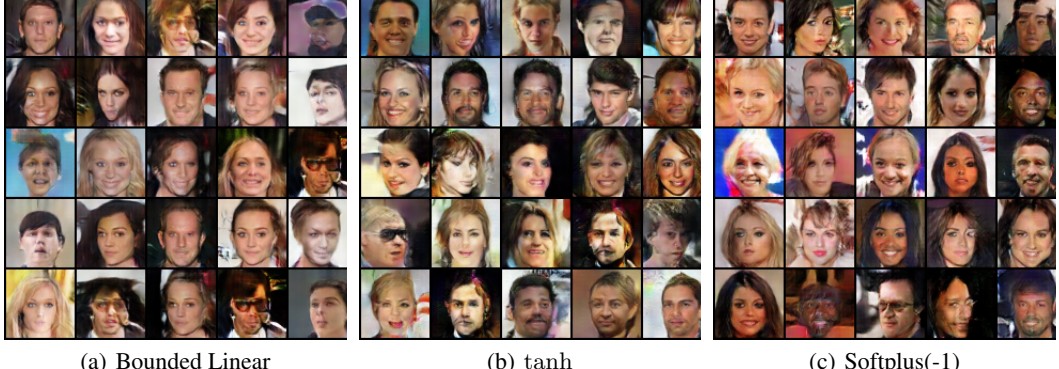

| (a) Bounded Linear | (b) tanh | (c) Softplus(-1) |

Figure 6: Generated images from the CelebA dataset. As observed the shape of faces are very well captured. Even softplus works well, albeit converges slower than bounded linear and tanh.

of colors we can expect to see in generators with weight clipping is significantly smaller. The final results for various other experiments are shown in Appendix C.

**LSUN Dataset**: Similar to the WGAN paper, we use our model to train on the LSUN bedrooms dataset Yu et al. (2015). LSUN dataset, compared to CIFAR-10, has more samples and are larger in size. We can generate larger images in this experiment, $64 \times 64$ and $128 \times 128$. Samples of the generated images are shown in Figure 5. Interestingly, our algorithm converges and produces good quality images at $40,000$th iteration. WGAN produces, with arguably worst quality, at iteration $60,000$. We observe, with batch-normalization linear bounded model is not as sharp as its alternative. Moreover, tanh does not show as much diversity in the generated images which we believe is due to the restricted nature of the output function.

**CelebA Dataset**: For the last experiment, we use the CelebA dataset Liu et al. (2015). We generate $64 \times 64$ images as shown in Figure 6. We use Softplus as another measure based on our framework where the divergence is interpreted as the difference of the log-likelihood of the observations from real and fake data (details are provided in Appendix C).

We observe even though the generated samples have good quality, Softplus converges slower than two other alternatives (takes almost twice the number of epochs).

## 6 CONCLUSION

In this paper we develop the building blocks of Lipschitz neural networks using which and Dudley metric, we propose a new GAN called Dudley GAN. Dudley GAN's discriminator is guaranteed to be a bounded Lipschitz neural network, thus produces state of the art result in both converge and sample quality. In addition, we show Dudley GAN and its extension is directly related to empirical risk minimization. This allows us to bridge the gap between GANs and learning complexity and generalization. We provide an upper bound on probability of confusion between real and fake images. We believe this view opens new avenues in GAN research for better understanding and quantifying the ability of the algorithm to capture the distribution of real samples.

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

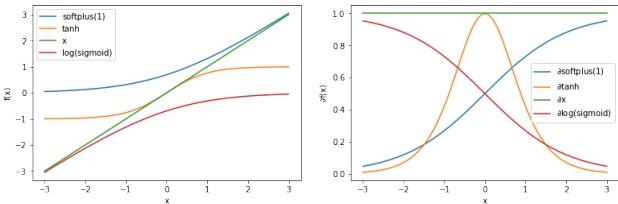

Figure 7: Various outputs for the neural net and their derivatives. For the original WGAN the derivative is constant, while the `tanh` has a gradient that decreases as the absolute value of the input increases.

## A  PROOFS

*proof of 1* (ii). For two Lipschitz continuous functions $f$ and $g$ with $L_f$ and $L_g$ constants we have:

$$\|f(g(\mathbf{x}_1)) - f(g(\mathbf{x}_2))\| \leq L_f \|g(\mathbf{x}_1) - g(\mathbf{x}_2)\| \leq L_f \times L_g \|\mathbf{x}_1 - \mathbf{x}_2\|$$

$\square$

*proof of 2*(i). Let $*$ denote convolutional operation between the input $\mathbf{x}$ and weight vector $\mathbf{w}$. Using Young's convolution inequality we have,

$$\|\mathbf{w} * \mathbf{x}_1 - \mathbf{w} * \mathbf{x}_2\| \leq |\mathbf{w}|\|\mathbf{x}_1 - \mathbf{x}_2\|,$$

To establish the bound for the $d$-dimensional weight on $|\mathbf{w}| = \sum_i |w_1| + \ldots, |w_d|$, we consider two cases (1) $|w_i| \leq 1$ where the elements are upper bounded by $d$; (2) $|w_i| > 1$ we know $w_1^2 + w_2^2 + \ldots, w_n^2 = C^2$, then $w_i^2 \leq C^2$ and $|w_i| \leq C$. Thus, $|\mathbf{w}| \leq Cd$. $\square$

*proof of 2 (ii).* We have $\|\langle \mathbf{w}, \mathbf{x}_1 \rangle - \langle \mathbf{w}, \mathbf{x}_2 \rangle\| \leq \|\mathbf{w}\|\|\mathbf{x}_1 - \mathbf{x}_2\|$ and $\|\mathbf{w}\| = C$. $\square$

*proof of 2(iii).* Sigmoid and `tanh` are continuously differentiable and their derivative is globally bounded, hence they are Lipschitz continuous. Considering all possible cases of the input to the ReLU function, it is easy to see $\|\text{ReLU}(x_1) - \text{ReLU}(x_2)\| \leq \|x_1 - x_2\|$. $\square$

*proof of 2*(iv). `log` is a differentiable function with bounded derivative in the specified domain. $\square$

*Proof of Equation 4.* Let's denote by $\mathcal{H}$ the heaviside function and $\mathcal{A} : \mathbb{R} \to [0, 1]$ as

$$\mathcal{A}(a) = \begin{cases} 1 & \text{if } a > 0 \\ 1 + \frac{a}{\gamma} & \text{if } 0 \leq a \leq \gamma \\ 0 & \text{otherwise} \end{cases} \tag{5}$$

From Theorem 4.9 in (Shawe-Taylor & Cristianini, 2004) and since $\mathcal{H} - 1 \leq \mathcal{A} - 1$ we have,

$$
\begin{aligned}
\mathbb{E}_{\mathcal{X} \cup \mathcal{X}'}[\mathcal{H}(-yg(\mathbf{x})) - 1] &\leq \mathbb{E}_{\mathcal{X} \cup \mathcal{X}'}[\mathcal{A}(-yg(\mathbf{x})) - 1] \\
&\leq \hat{\mathbb{E}}_{\mathcal{X} \cup \mathcal{X}'}[\mathcal{A}(-yg(\mathbf{x})) - 1] + \mathfrak{R} + 3\sqrt{\frac{\log(2/\delta)}{2m}}
\end{aligned} \tag{6}
$$

and since $\mathcal{A}(-y_i f(\mathbf{x}_i)) \leq \frac{(yf(\mathbf{x}) - \gamma)_+}{\gamma}$ for $i = 1, \ldots, m$, we have

$$\mathbb{E}_{\mathcal{X} \cup \mathcal{X}'}[\mathcal{H}(-yg(\mathbf{x})) - 1] \leq \frac{1}{m\gamma}\sum_{i=1}^{m}(yf(\mathbf{x}) - \gamma)_+ + \mathfrak{R} + 3\sqrt{\frac{\log(2/\delta)}{2m}} \tag{7}$$

and since the support of the distribution is in the ball centered on the origin, we have the inequality proven. $\square$

## B  NETWORK ARCHITECTURE

We have implemented our approach in PyTorch.

| | |
|---|---|
| Weight-normalized-Conv2d | (nc, 32, kernel size=(4, 4), stride=(2, 2), padding=(1, 1)) |
| LeakyReLU | (0.2, inplace) |
| Weight-normalized-Conv2d | (32, 64, kernel size=(4, 4), stride=(2, 2), padding=(1, 1)) |
| LeakyReLU | (0.2, inplace) |
| Weight-normalized-Conv2d | (64, 128, kernel size=(4, 4), stride=(2, 2), padding=(1, 1)) |
| LeakyReLU | (0.2, inplace) |
| Weight-normalized-Conv2d | (128, 256, kernel size=(4, 4), stride=(2, 2), padding=(1, 1)) |
| LeakyReLU | (0.2, inplace) |
| Weight-normalized-Conv2d | (256, 100, kernel size=(4, 4), stride=(1, 1)) |
| Linear | (100, 1000) |
| ReLU | () |
| Linear | (1000, 1) |

Table 2: The discriminator for image size $32 \times 32$. $nc$ is the number of channels which is 1 for MNIST and 3 for colored images.

| | |
|---|---|
| ConvTranspose2d | (100, 256, kernel size=(4, 4), stride=(1, 1)) |
| BatchNorm2d | (256) |
| ReLU | () |
| ConvTranspose2d | (256, 128, kernel size=(4, 4), stride=(2, 2), padding=(1, 1)) |
| BatchNorm2d | (128) |
| ReLU | () |
| ConvTranspose2d | (128, 64, kernel size=(4, 4), stride=(2, 2), padding=(1, 1)) |
| BatchNorm2d | (64) |
| ReLU | () |
| ConvTranspose2d | (64, 32, kernel size=(4, 4), stride=(2, 2), padding=(1, 1)) |
| BatchNorm2d | (32) |
| ReLU | () |
| ConvTranspose2d | (32, nc, kernel size=(4, 4), stride=(2, 2), padding=(1, 1)) |
| Tanh | () |

Table 3: The generator for image size $32 \times 32$. $nc$ is the number of channels which is 1 for MNIST and 3 for colored images.

## C  LOG-LIKELIHOOD DIFFERENCE

From Proposition (iv), it is clear that the Softplus function satisfies Lipschitz condition for $\beta \neq 0$. In particular, for $\beta = -1$, we recover log of a sigmoid function which is in par with the original GAN except instead of $\mathbb{E}_{\mathbf{x} \sim p}[\log(D_\mathbf{w}(\mathbf{x}))] + \mathbb{E}_{\mathbf{z} \sim p_\mathbf{z}}[\log(1 - D_\mathbf{w}(g(\mathbf{z})))]$ we train

$$\mathbb{E}_{\mathbf{x} \sim p}[\log(D_\mathbf{w}(\mathbf{x}))] - \mathbb{E}_{\mathbf{z} \sim p_\mathbf{z}}[\log(D_\mathbf{w}(g(\mathbf{z})))]$$

for a sigmoid function $D_\mathbf{w}$ (note $f_\mathbf{w}(\mathbf{x}) = \log(D_\mathbf{w}(\mathbf{x}))$ in Equation 1) with $-\log(D(\mathbf{x}))$ trick for the generator. This choice of $f_\mathbf{w}$ allows us to interpret our GAN as minimizing the difference of the expected log-likelihood in the discriminator (as a divergence) while maximizing the expected log likelihood in the generator. For the case when we use Softplus($-1$) function we employ the same logic and add the regularizer to encourage the function values to be close to boundary as $\log(0.5)$ which yields a bounded function.

## ADDITIONAL EXPERIMENTS

**CIFAR-10 Dataset**:

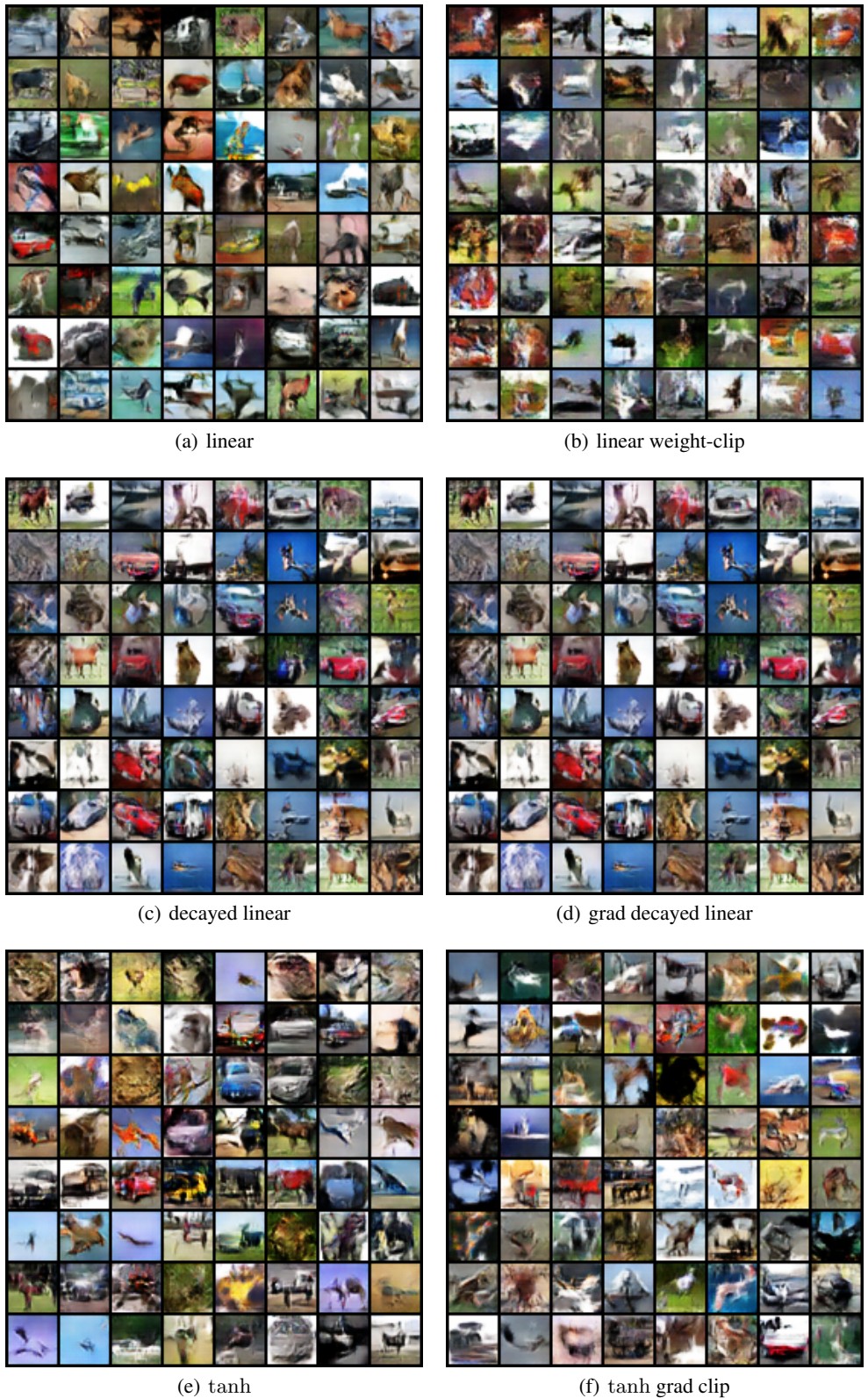

Figure 8: Figure

**LSUN Dataset**:

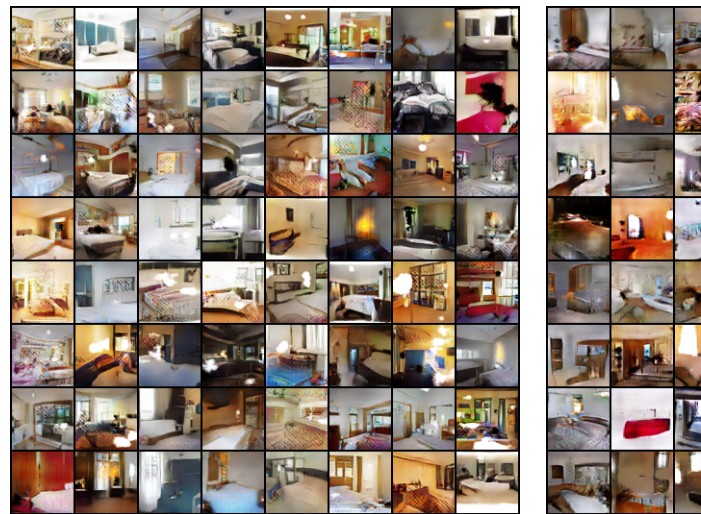
(a) Linear with Batch-norm

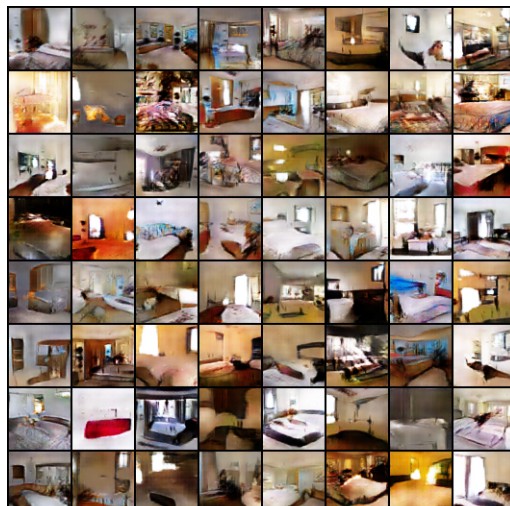
(b) No Batch-norm

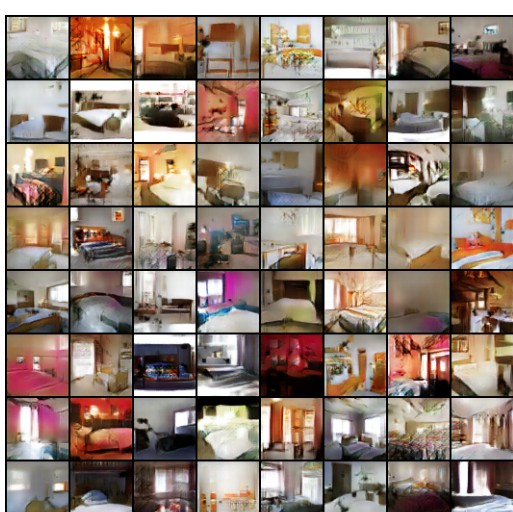
(c) tanh

Figure 9: LSUN Images

**CelebA Dataset**:

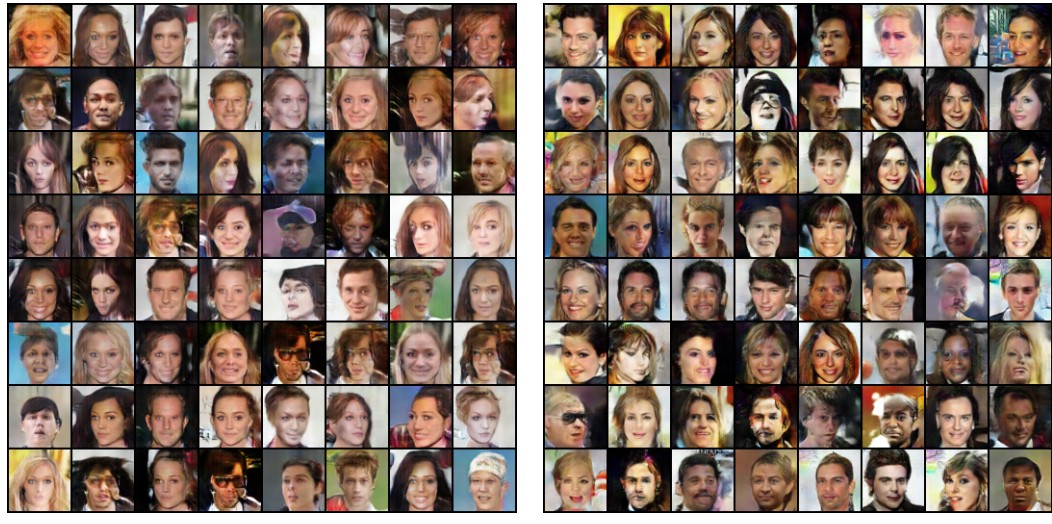

(a) Linear (b) tanh

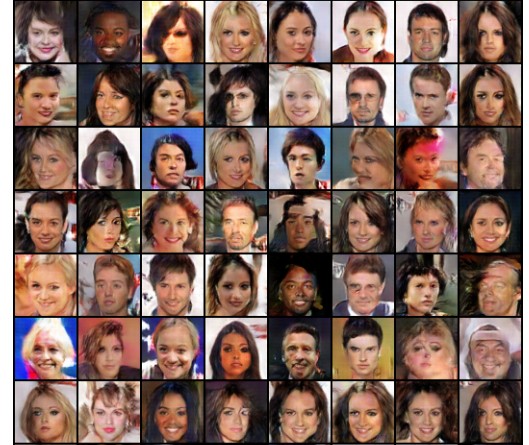

(c) Softplus(-1)

