# OpenReview forum: "Deep Lipschitz networks and Dudley GANs"
_ICLR.cc/2018/Conference — Reject_

### Official Review · AnonReviewer2 · 2017-11-16
**There are certain contributions in literatures, but the novelty may be not significant.**

**Rating:** 8
**Confidence:** 4

**Review:**

Ensuring Lipschitz condition in neural nets is essential of stablizing GANs. This paper proposes two contraint-based optimzation to ensure the Lips condtions , and these proposed approaches maintain suffcient capacity, as well as expressiveness of the network.  A simple theoritical result is given by emprical risk minimization. The content of this paper
is written clearly, and there are certain contribution and orginality in the literature. However, I am not sure that the novelty is
significant, since I think that the idea of proposing their optimization is trival. Here I am concerned with the following two questions:
(1) How to parameterize the function space of f_w or h_w, since they are both multivariate and capacity of the network will be
reduced if the used way of parametering functions is adopted inappropriatily.
(2) The theoretical result in (4)  doesnot contain the information of Rademacher complexity, and it may be suboptimal in some sense. Besides, the parameter $\gamma$ appears in the discriminator, which contradicts its role on the contraint of functions space.

---

> ### Author Response · Authors · 2017-12-17
> **Please see the response below**
>
> Thanks for the review. For (1) we should note all the comparisons are for the same parametric function (a neural network as the universal estimator for the generator/discriminator). If there are not enough parameters in the network, the capacity of the network won’t be enough to learn an appropriate discriminator/generator. In the same scenario our approach provides more capacity and better expressiveness for the discriminator. As shown in Figure 3, this leads to larger variance (higher diversity) in the generated samples.
> For (2), the second term in Equation 4 is an upper bound on the Rademacher complexity. Instead if Rademacher bound is used in Equation 4, the bound becomes tighter (Rademacher complexity is intractable in practice). Thanks for pointing out the typo in reference to $\gamma$: it is (\gamma-yf(x)) in Equation 4.

---

### Official Review · AnonReviewer1 · 2017-12-01
**Well-written, but unlikely a big step forward for GANs**

**Rating:** 5
**Confidence:** 3

**Review:**

The authors propose a different type of GAN--the Dudley GAN--that is related to the Dudley metric. In fact, it is very much like the WGAN, but rather than just imposing the function class to have a bounded gradient, they also impose it to be bounded itself. This is argued to be more stable than the WGAN, as gradient clipping is said not necessary for the Dudley GAN. The authors empirically show that the Dudley GAN achieves a greater LL than WGAN for the MNIST and CIFAR-10 datasets.

The main idea [and its variants] looks solid, but with the plethora of GANs in the literature now, after reading I'm still left wondering why this GAN is significantly better than others [BEGAN, WGAN, etc.]. It is clear that imposing the quadratic penalty in equation (3) is really the same constraint as the Dudley norm? The big contribution of the paper seems to be that adding some L_inf regularization to the function class helps preclude gradient clipping, but after reading I'm unsure why this is "the right thing" to do in this case. We know that convergence in the Wasserstein metric is stronger than the Dudley metric, so why is using the weaker metric overweighed by the benefits in training?

Nits: Since the function class is parameterized by a NN, the IPM is not actually the Dudley metric between the two distributions. One would have to show that the NN is dense in Dudley unit ball w.r.t. L_inf norm, but this sort of misnaming had started with the "Wasserstein" GAN.

---

> ### Author Response · Authors · 2017-12-17
> **Weak convergence is important for GANs. Please see the complete response below.**
>
> Thanks for the review. Firstly, we believe we provided the tools for building variants of IPM-based GANs. We believe our approach is simple to implement and understand and provided better capacity in the discriminator so that the generator can produce better images. We argue that weight clipping is definitely not the best way to approach Lipschitz condition in neural networks (something that authors point out in WGAN paper too). Having a bounded output also allows us to analyse the model from the complexity theory point of view as well—something that has never been investigated before in the GAN community. With Dudley metric, we need the Lipchitz function to be bounded and the norm introduced by it to be smaller than a constant.
> Since we use a Lipschitz neural network and use the (soft) penalty to ensure the function value remains bounded we believe we have the correct norm. This norm however is not a constant value throughout training.
> Moreover, we believe our contributions are beyond just the regularization: we showed a simple regularization combined with principles for building a Lipschitz neural network can overcome the problems with quality, speed of convergence and expressiveness of WGAN. We should use a bounded continuous function, we can draw parallels to complexity theory and show how GANs can be interpreted as ERM with a bound on the probability of discrimination between real and fake samples. We also showed for the same divergence measure and same neural net structure, simple change in the output of the network will change the quality of samples and convergence properties (tanh is slower but can be more stable).
> We agree that the convergence in Dudley is weaker than the Wasserstein metric, however in the context of GANs this is not a concern because we learn the generator and we need to think of the interplay between the generator we learn and the divergence we consider. With GANs, since we learn the generator functions as well strong convergence is not necessarily what we are interested in. It is already shown that the weak convergence for GANs is more favourable because it stops the discriminator from saturation*. In addition, with GANs the convergence is not calculated using the expectation of the function with respect to the true measure (we only use a subset of observations to estimate the empirical mean). As such, the weaker convergence does not necessarily imply poorer performance in practice. There are other factors that have to be considered to have a stable training that converges and produces real looking samples. In addition, for the WGAN, the weights are extremely limited that will cause slow convergence. In our experiments, we observe faster convergence compared to WGAN. As an example, total variation which provides strong convergence has not been very successful in its applications to GANs. We believe the constrains we introduced to the discriminator in Dudley GAN is sufficient enough to deter the discriminator from fast convergence while being expressive enough to capture the complexity of the data (unlike weight clipping in WGAN).
> *: Approximation and Convergence Properties of Generative Adversarial Learning: https://arxiv.org/pdf/1705.08991.pdf

---

### Official Review · AnonReviewer3 · 2017-12-01
**Marginally below acceptance threshold**

**Rating:** 5
**Confidence:** 1

**Review:**

It is clear that the problem studied in this paper is interesting. However, after reading through the manuscript, it is not clear to me what are the real contributions made in this paper. I also failed to find any rigorous results on generalization bounds. In this case, I cannot recommend the acceptance of this paper.

---

> ### Author Response · Authors · 2017-12-17
> **Please explain where the contributions are not clear. Please see the complete response below.**
>
> Thanks for the review. In this paper, we provided the necessary conditions for a Lipschitz neural nets which have applications beyond GANs. In addition, we believe we provided the tools for building variants of IPM-based GANs. We believe our approach is simple to implement and understand and provided better capacity in the discriminator so that the generator can produce better images. In addition, this Lipschitz neural network can be used in WGANs instead of weight clipping. Weight-clipping significantly limits the capacity of the network which leads to worst quality for the samples generated using WGAN. For GANs the discriminator has to be controlled so that it improves gradually. If it learns to discriminate very fast, then the generator will not be able to produce good quality images. We propose to use L_inf norm corresponding to using Dudley metric in the discriminator (instead of just limiting the weights). This regularization in addition to the Lipschitz neural network we introduced allows for better quality images to be produced faster compared to WGAN. We believe, the divergence measure we employed in our paper is more suitable for GANs than other alternatives. It is expressive that allows to learn good generators and not too restrictive to decrease the convergence speed.
> Moreover, since we use bounded functions, we can devise the generalization bound in the paper. This bound (that has never been studied or used before) suggests the relation between the discriminator’s performance with respect to the margin and the input bound.
> We appreciate it if the reviewer could provide more information on which part of our contributions are not clear enough so that we provide better explanation.

---

### Decision · Program_Chairs · 2018-01-29
**ICLR 2018 Conference Acceptance Decision**

**Decision:**

Reject

**Comment:**

Dear authors,

While the reviewers appreciated your analysis, they all expressed concerns about the significance of the paper. Indeed, given the plethora of GAN variants, it would have been good to get stronger evidence about the advantages of the Dudley GAN. Even though I agree it is difficult to provide a clean comparison between generative models because of the lack of clear objectives, the LL on one dataset and images generated is limited. For instance, it would have been nice to show robustness results as this is a clear issue with GANs.